# Hierarchical Modular Optimization of Convolutional Networks Achieves Representations Similar to Macaque IT and Human Ventral Stream

**Daniel Yamins**[*]
McGovern Institute of Brain Research
Massachusetts Institute of Technology
Cambridge, MA 02139
yamins@mit.edu

**Ha Hong**[*]
McGovern Institute of Brain Research
Massachusetts Institute of Technology
Cambridge, MA 02139
hahong@mit.edu

**Charles Cadieu**
McGovern Institute of Brain Research
Massachusetts Institute of Technology
Cambridge, MA 02139
hahong@mit.edu

**James J. Dicarlo**
McGovern Institute of Brain Research
Massachusetts Institute of Technology
Cambridge, MA 02139
dicarlo@mit.edu

## Abstract

Humans recognize visually-presented objects rapidly and accurately. To understand this ability, we seek to construct models of the ventral stream, the series of cortical areas thought to subserve object recognition. One tool to assess the quality of a model of the ventral stream is the Representational Dissimilarity Matrix (RDM), which uses a set of visual stimuli and measures the distances produced in either the brain (i.e. fMRI voxel responses, neural firing rates) or in models (features). Previous work has shown that all known models of the ventral stream fail to capture the RDM pattern observed in either IT cortex, the highest ventral area, or in the human ventral stream. In this work, we construct models of the ventral stream using a novel optimization procedure for category-level object recognition problems, and produce RDMs resembling both macaque IT and human ventral stream. The model, while novel in the optimization procedure, further develops a long-standing functional hypothesis that the ventral visual stream is a hierarchically arranged series of processing stages optimized for visual object recognition.

## 1 Introduction

Humans recognize visually-presented objects rapidly and accurately even under image distortions and variations that make this a computationally challenging problem [27]. There is substantial evidence that the human brain solves this invariant object recognition challenge via a hierarchical cortical neuronal network called the ventral visual stream [13, 17], which has highly homologous areas in non-human primates [19, 9]. A core, long-standing hypothesis is that the visual input captured by the retina is rapidly processed through the ventral stream into an effective, "invariant" representation of object shape and identity [11, 9, 8]. This hypothesis has been bolstered by recent developments in neuroscience which have shown that abstract category-level visual information is accessible in IT (inferotemporal) cortex, the highest ventral cortical area, but much less effectively accessible in lower areas such as V1, V2 or V4 [23]. This observation has been confirmed both at the individual neural level, where single-unit responses can be decoded using linear classifiers

---

[*] web.mit.edu/ yamins; [*] These authors contributed equally to this work.

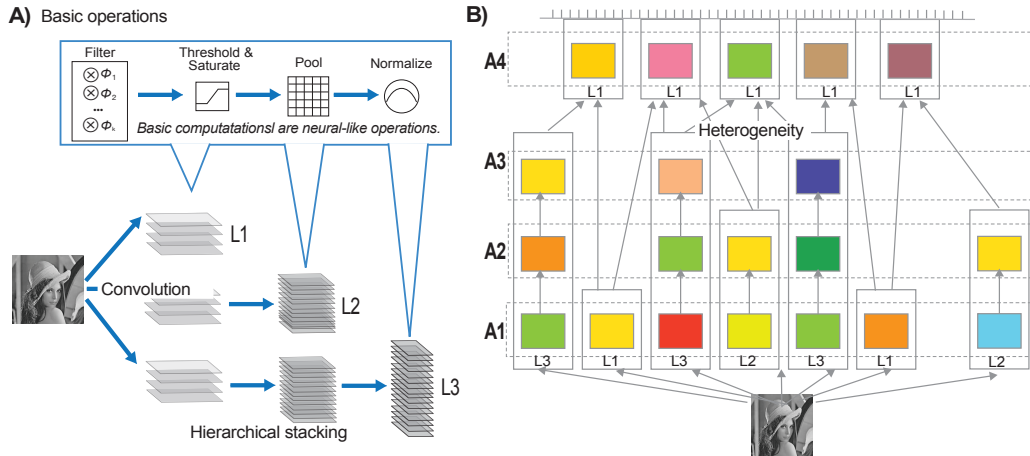

Figure 1: A) Heterogenous hierarchical convolutional neural networks are composed of **basic operations** that are simple and neurally plausibly, including linear reweighting (filtering), thresholding, pooling and normalization. These simple elements are **convolutional** and are **stacked hierarchically** to construct non-linear computations of increasingly greater power, ranging through low (L1), medium (L2), and high (L3) complexity structures. B) Several of these elements are combined to produce mixtures capturing **heterogenous** neural populations. Each processing stage across the heterogeneous networks (A1, A2, ...) can be considered an analogous to a neural visual area.

to to yield category predictions [14, 23] and at the population code level, where response vector correlation matrices evidence clear semantic structure [19].

Developing *encoding models*, models that map the stimulus to the neural response, of visual area IT would likely help us to understand object recognition in humans. Encoding models of lower-level visual responses (RGC, LGN, V1, V2) have been relatively successful [21, 4] (but cf. [26]). In higher-visual areas, particularly IT, theoretical work has described a compelling framework which we ascribe to in this work [29]. However, to this point it has not been possible to produce effective encoding models of IT. This explanatory gap, between model responses and IT responses, is present at both the level of the individual neuron responses and at the population code level. Of particular interest for our analysis in this paper, current models of IT, such as HMAX, have been shown to fail to achieve the specific categorical structures present in neural populations [18]. In other related work, descriptions of higher areas (V4, IT) responses have been made for very narrow classes of artificial stimuli and do not define responses to arbitrary natural images [6, 3].

In a step toward bridging this explanatory gap, we describe advances in constructing models that capture the categorical structures present in IT neural populations and fMRI measurements of humans. We take a top-down functional approach focused on building invariant object representations, optimizing biologically-plausible computational architectures for high performance on a challenging object recognition screening task. We then show that these models capture key response properties of IT, both at the level of individual neuronal responses as well as the neuronal population code – even for entirely new objects and categories never seen in model selection.

## 2 Methods

### 2.1 Heterogenous Hierarchical Convolutional Models

Inspired by previous neuronal modeling work [7, 6], we constructed a model class based on three basic principles: (i) single layers composed of neurally-plausible **basic operations**, including filtering, nonlinear activation, pooling and normalization (ii) using **hierarchical stacking** to construct more complex operations, and (iii) **convolutional weight sharing** (fig. 1A). This general type of model has been successful in describing a variety of phenomenology throughout the ventral stream [30]. In addition, we allow combinations of multiple hierarchical components each with different

parameters (such as pooling size, number of filters, etc.), representing different types of units with different response properties [5] and refer to this concept as (iv) **heterogeneity** (fig. 1B).

We will now formally define the class of heterogeneous hierarchical convolutional neural networks, $\mathcal{N}$. First consider a simple neural network function defined by

$$N_\Theta = Pool_{\theta_p}(Normalize_{\theta_N}(Threshold_{\theta_T}(Filter_{\theta_F}(\textit{Input})))) \qquad (1)$$

where the pooling, normalization, thresholding and filterbank convolution operations are as described in [28]. The parameters $\Theta = (\theta_p, \theta_N, \theta_T, \theta_F)$ control the structure of the constituent operations. Each model stage therefore actually represents a large family of possible operations, specified by a set of parameters controlling e.g. fan-in, activation thresholds, pooling exponents, spatial interaction radii, and template structure. Like [28], we use randomly chosen filterbank templates in all models, but additionally allow the mean and variance of the filterbank to vary as parameters. To produce deep feedforward networks, single layers are stacked:

$$\mathbf{P}_{\Theta_{P,l-1}}^{\ell-1} \xrightarrow{Filter} \mathbf{F}_{\Theta_{F,l}}^\ell \xrightarrow{Threshold} \mathbf{T}_{\Theta_{T,l}}^\ell \xrightarrow{Normalize} \mathbf{N}_{\Theta_{N,l}}^\ell \xrightarrow{Pool} \mathbf{P}_{\Theta_{P,l}}^\ell \qquad (2)$$

We denote such a stacking operation as $N(\Theta_1, \ldots, \Theta_k)$, where the $\Theta_l$ are parameters chosen separately for each layer, and will refer to networks of this fork as "single-stack" networks. Let the set of all depth-$k$ single-stack networks be denoted $\mathbb{N}_k$. Given a sequence of such single-stack networks $\mathbf{N}(\Theta_{i1}, \Theta_{12}, \ldots, \Theta_{in_i})$ (possibly of different depths), the combination $\mathbb{N} \equiv \oplus_{i=1}^k \mathbf{N}(\Theta_{i1}, \Theta_{12}, \ldots, \Theta_{in_i})$ is formed by aligning the output layers of these models along the spatial convolutional dimension. These networks $\mathbb{N}$ can, of course, also be stacked, just like their single-stack constituents, to form more complicated, deeper heterogenous hierarchies. By definition, the class $\mathcal{N}$ consists of all the iterative chainings and combinations of such networks.

## 2.2 High-Throughput Screening via Hierarchical Modular Optimization

Our goal is to find models within $\mathcal{N}$ that are effective at modeling neural responses to a wide variety of images. To do this, our basic strategy is to perform high-throughput optimization on a screening task [28]. By choosing a screening task that is sufficiently representative of the aspects that make the object recognition problem challenging, we should be able to find network architectures that are generally applicable. For our screening set, we created a set of 4500 synthetic images composed of 125 images each containing one of 36 three-dimensional mesh models of everyday objects, placed on naturalistic backgrounds. The screening task we evaluated was 36-way object recognition. We trained Maximum Correlation Classifiers (MCC) with 3-fold cross-validated 50%/50% train/test splits, using testing classification percent-correct as the screening objective function.

Because $\mathcal{N}$ is a very large space, determining among the vast space of possibilities which parameter setting(s) produce visual representations that are high performing on the screening set, is a challenge. We addressed this by applying a novel method we call Hierarchical Modular Optimization (HMO). The intuitive idea of the HMO optimization procedure is that a good multi-stack heterogeneous network will be found by creating mixtures of single-stack components each of which specializes in a portion of an overall problem. To achieve this, we implemented a version of adaptive hyperparameter boosting, in which rounds of optimization are interleaved with boosting and hierarchical stacking.

Specifically, suppose that $N \in \mathcal{N}$ and $S$ is a screening stimulus set. Let $E$ be the binary-valued classification correctness indicator, assigning to each stimulus image $s$ 1 or 0 according to whether the screening task prediction was right or wrong. Let $score(N, S) = \sum_{s \in S} N(F(s))$. To efficiently find $N$ that maximizes $score(N, S)$, the HMO procedure follows these steps:

**1. Optimization:** Optimize the score function within the class of single-stack networks, obtaining an optimization trajectory of networks in $\mathcal{N}$ (fig 2A, left). The optimization procedure that we use is Hyperparameter Tree Parzen Estimator, as described in [1]. This procedure is effective in large parameter spaces that include discrete and continuous parameters.

**2. Boosting:** Consider the set of networks explored during step 1 as a set of weak learners, and apply a standard boosting algorithm (Adaboost) to identify some number of networks $N_{11}, \ldots, N_{1l_1}$ whose error patterns are complementary (fig 2A, right).

**3. Combination:** Form the multi-stack network $N_1 = \oplus_i N_{1i}$ and evaluate $E(N_1(s))$ for all $s \in S$.

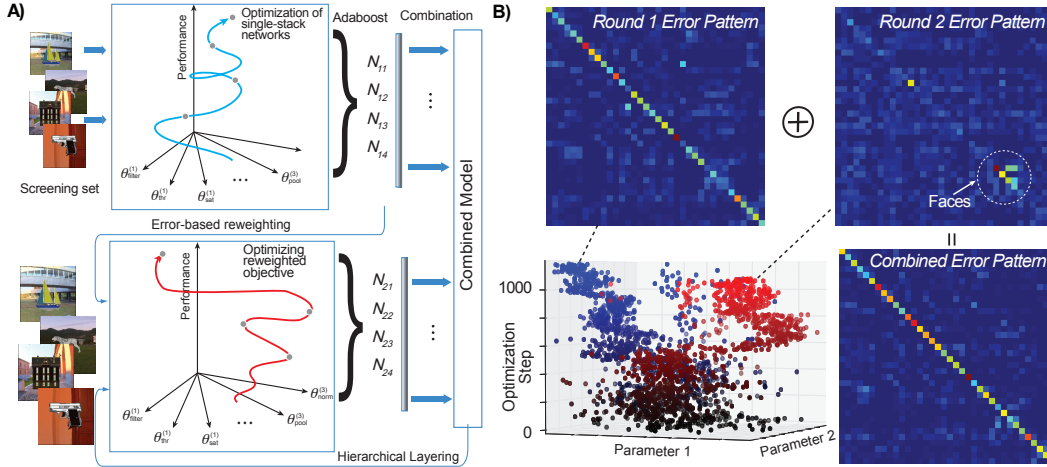

Figure 2: A) The Hierarchical Modular Optimization is a mechanism for efficiently optimizing neural networks for object recognition performance. The intuitive idea of HMO is that a good multi-stack hetergenous network will be found by creating mixtures of single-stack components each of which specializes in a portion of an overall problem. The process first identifies complementary performance gradients in the space of single-stack (non-heterogenous) convolutional neural networks by using version of adaptive boosting interleaved with hyperparameter optimization. The components identified in this process are then composed nonlinearly using a second convolutional layer to produce a combined output model. B) Top: the 36-way confusion matrices associated with two complementary components identified in the HMO process. Bottom Left: The two optimization trajectories from which the single-stack models were drawn that produced the confusion matrices in the top panels. The optimization criterion for the second round (red dots) was defined relative to the errors of the first round (blue dots). Bottom Right: The confusion matrix of the heterogenous model produced by combining the round 1 and round 2 networks.

**4. Error-based Reweighting:** Repeat step 1, but reweight the scoring to give the $j$-th stimulus $s_j$ weight 0 if $N_1$ is correct in $s_j$, and 1 otherwise. That is, the performance function to be optimized for $N$ is now $\sum_{s \in S} E(N_1(s)) \cdot E(N(s))$. Repeat the step 2 on the results of the optimization trajectory obtained to get models $N_{21}, \ldots N_{2k_2}$, and repeat step 3. Steps 1, 2, 3 are repeated $K$ times.

After $K$ repeats, we will have obtained a multi-stack network $N = \oplus_{i \leq K, j \leq k_i} N_{ij}$. The process can then simply be terminated, or repeated with the output of $N$ as the input to another stacked network. In the latter case, the next layer is chosen using the same model class $\mathcal{N}$ to draw from, and using the same adaptive hyperparameter boosting procedure.

The meta-parameters of the HMO procedure include the numbers of components $l_1, l_2, \ldots$ to be selected at each boosting round, the number of times $K$ that the interleaved boosting and optimization is repeated and the number of times $M$ this procedure is stacked. To constrain this space we fix the metaparameters $l_1 = l_2 \ldots . . = 10$, $K = 3$, and $M \leq 2$. With the fixed screening set described above, and these metaparameter settings, we generated a network $N_{HMO}$. We will refer back to this model throughout the rest of the paper. $N_{HMO}$ produces 1250-dimensional feature vectors for any input stimulus; we will denote $N_{HMO}(s)$ as the resulting feature vector for stimulus $s$ and $N_{HMO}(s)_k$ as its $k$-th component in 1250-dimensional space.

## 2.3 Predicting IT Neural Responses

Utilizing the $N_{HMO}$ network, we construct models of IT in one of two ways: 1) we estimate a GLM model predicting individual neural responses or 2) we estimate linear classifiers of object categories to produce a candidate IT neural space.

To construct models of individual neural responses we estimate a linear mapping from a non-linear space produced by a model. This procedure is a standard GLM of individual neural responses. Because IT responses are highly non-linear functions of the input image, successful models must

capture the non-linearity of the IT response. The $N_{HMO}$ network produces a highly-nonlinear transformation of the input image and we compare the efficacy of this non-linearity against those produced by other models. Specifically for a neuron $n_i$, we estimate a vector $w_i$ to minimize the regression error from $N_{HMO}$ features to $n_i$'s responses, over a training set of stimuli. We evaluate goodness of fit of by measuring the regression $r^2$ values between the neural response and the GLM predictions on held-out images, averaged over several train/test splits. Taken over the set of predicted neurons $n_1, n_2, ... n_k$, the collection of regression weight vectors $\mathbf{w}_i$ comprise a matrix $W$ that can be thought of as a final linear top level that forms part of the model of IT. This method evidently requires the presence of low-level neural data on which to train.

We also produce a candidate IT neural space by estimating linear classifiers on an object recognition task. As we might expect different subregions of IT cortex to have different selectivities for object categories (for example face, body, and place patches [15, 10]), the output of the linear classifiers will also respond preferentially to different object categories. We may be able to leverage some understanding of what a subregion's task specialization might be to produce the weighting matrix $W$. Specifically, we estimate a linear mapping $W$ to be the weights of a set of linear classifiers trained from the $N_{HMO}$ features on a specific set of object recognition tasks. We can then evaluate this mapping on a novel set of images and compare to measured IT or human ventral stream data. This method may have traction even when individual neural response data are not available.

### 2.4 Representational Dissimilarity Matrices

Implicit in this discussion is the idea of comparing two different representations (in this case, the model's predicted population versus the real neural population) on a fixed stimulus set. The Representational Dissimilarity Matrix (RDM) is a convenient tool for this comparison [19]. Formally, given a stimulus set $S = s_1, \ldots, s_k$ and vectors of neural population responses $R = \vec{r}_1, \ldots, \vec{r}_k$ in which $r_{ij}$ is the response of the $j$-th neuron to the $i$-th stimulus, define

$$RDM(R)_{ij} = 1 - \frac{cov(r_i, r_j)}{var(r_i) \cdot var(r_j)}.$$

The $RDM$ characterizes the layout of the stimuli in high-dimensional neural population space. Following [19], we measured similarity between population representations as the Spearman rank correlations between the RDMs for two populations, in which both RDMs are treated as vectors in $k(k-1)/2$-dimensional space. Two populations can have similar RDMs on a given stimulus set, even if details of the neural responses are different.

## 3 Results

To test the $N_{HMO}$ model, we took two routes, corresponding to the two methods for prediction described above. First (sec. 3.1), we obtained our own neural data on a testing set of our own design and tested the $N_{HMO}$ model's ability to predict individual-level neural responses using the linear regression methodology described above. This approach allowed us to directly test the $N_{HMO}$ models' power in a setting were we had acess to low-level neural information. Second (sec. 3.2), we also compared to neural data collected by a different group, but only released at a very coarse level of detail – the RDMs of their measured population. This constraint required us to additionally posit a task blend, and to make the comparison at the population RDM level.

### 3.1 The Neural Representation Benchmark Image Set

We analyzed neural data collected on the Neural Representation Benchmark (NRB) dataset, which was originally developed to compare monkey neural and human behavioral responses [23, 2]. The NRB dataset consists of 5760 images of 64 distinct objects. The objects come from eight "basic" categories (animals, boats, cars, chairs, faces, fruits, planes, tables), with eight exemplars per category (e.g., BMW, Z3, Ford, &c for cars) (see fig 3B bottom left), with objects varying in position, size, and 3d-pose, and placed on a variety uncorrelated natural backgrounds. These parameters were varied concomitantly, picked randomly from a uniform ranges at three levels of object identity-preserving variation (low, medium, and high). The NRB set was designed to test explicitly the transformations of pose, position and size that are at the crux of the invariant object recognition problem. None of the

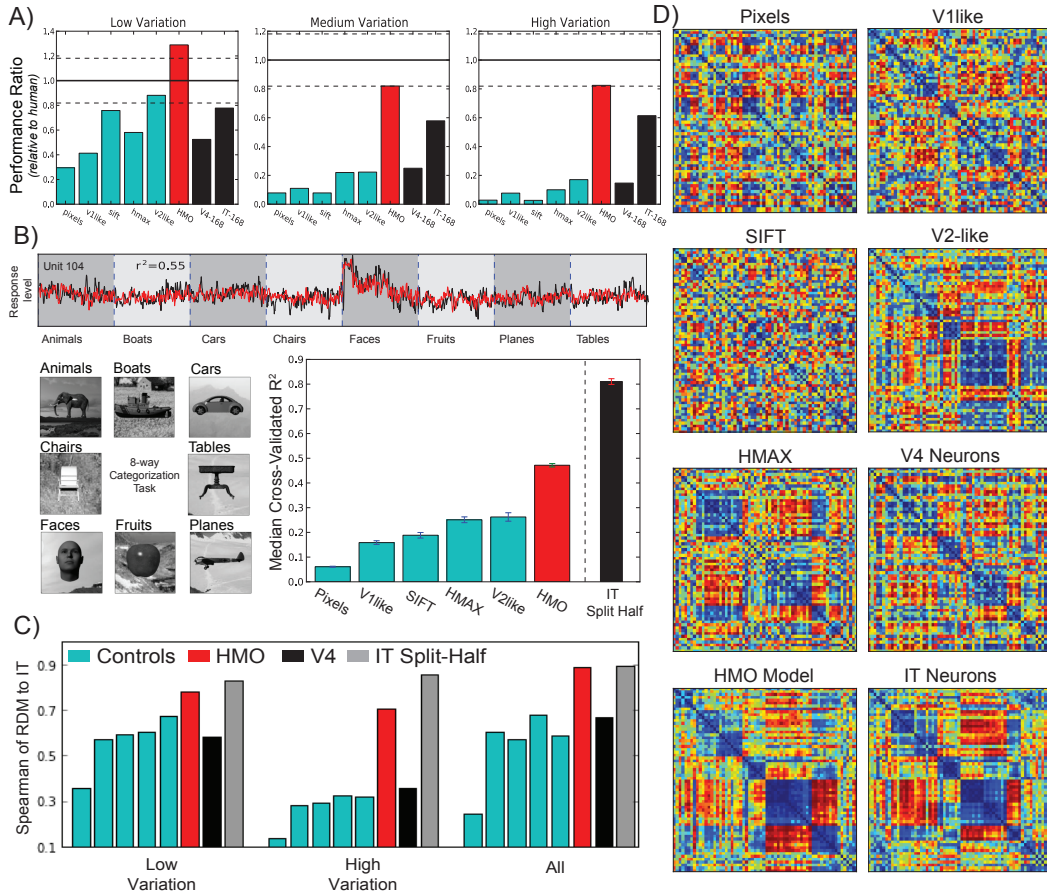

Figure 3: A) 8-way categorization performances. Comparison was made between several existing models from the literature (cyan bars), the HMO model features, and data from V4 and IT neural populations. Performances are normalized relative to human behavioral data collected from Amazon Mechanical Turk experiments. High levels variation strongly separates the HMO model and the high-level IT neural features from the other representations. B) Top: Actual neural response (black) vs. prediction (red) for a single sample IT unit. This neuron shows high functional selectivity for faces, which is effectively mirrored by the predicted unit. Bottom Left: Sample Neural Representation Benchmark images. C) Comparison of Representational Dissimilarity Matrices (RDMs) for NRB dataset. D) As populations increase in complexity and abstraction power, they become progressively more like that of IT, as category structure that was blurred out at lower levels by variability becomes abstracted at the higher levels. The HMO model shows similarity to IT both on the block diagonal structure associated with categorization performance, but also on the off-diagonal comparisons that characterize the neural representation more precisely.

objects, categories or backgrounds used in the HMO screening set appeared in the NRB set; moreover, the NRB image set was created with different image and lighting parameters, with different rendering software.

Neural data was obtained via large-scale parallel array electrophysiology recordings in the visual cortex of awake behaving macaques. Testing set images were presented foveally (central 10 deg) with a Rapid Serial Visual Presentation (RSVP) paradigm, involving passively viewing animals shown random stimulus sequences with durations comparable to those in natural primate fixations (e.g. 200 ms). Electrode arrays were surgically implanted in V4 and IT, and recordings took place daily over a period of several months. A total of 296 multi-unit responses were recorded from two animals. For each testing stimulus and neuron, final neuron output responses were obtained by averaging data from between 25 and 50 repeated trials. With this dataset, we addressed two questions: how well the HMO model was able to perform on the categorization tasks supported by the dataset, how well the HMO predicted the neural data.

Performance was assessed for three types of tasks, including 8-way basic category classification, 8-way car object identification, and 8-way face object identification. We computed the model's predicted outputs in response to each of the testing images, and then tested simple, cross-validated linear classifiers based on these features. As performance controls, we also computed features on the test images for a number of models from the literature, including a V1-like model [27], a V2-like model [12], and an HMAX variant [25]. We also compared to a simple computer vision model, SIFT[22], as well as the basic pixel control. Performances were also measured for neural output features, building on previous results showing that V4 neurons performed less well than IT neurons at higher variation levels[23], and confirming that the testing tasks meaningfully engaged higher-level vision processing. Figure 3A) compares overall performances, showing that the HMO-selected model is able to achieve human-level performance at all levels of variation. Critically, the HMO model performs well not just in low-variation settings in which simple lower-level models can do so, but is able to achieve near-human performance (within 1 std of the average human) even when faced with large amounts of variation which caused the other models to perform near chance. Since the testing set contains entirely different objects in non-overlapping basic categories, with none of the same backgrounds, this suggests that the nonlinearity identified in the HMO screening phase is able to achieve significant generalization across image domains.

Given that the model evidenced high transferable performance, we next determined the ability of the model to explain low-level neuronal responses using regression. The HMO model is able to predict approximately 48% of the explainable variance in the neural data, more than twice as much as any competing model (fig. 3B). Using the same transformation matrices $W$ obtained from the regression fitting, we also computed RDMs, which show significant similarity to IT populations at both nearly comparable to the split-half similarity range of the IT population itself (fig. 3C). A key comparison between models and data shows that as populations ascend the ventral hierarchy and increase in complexity, they become progressively closer to IT, with category structure that was blurred out at lower levels by variation becoming properly abstracted away at the higher levels (fig. 3D).

## 3.2    The Monkeys & Man Image Set

Kriegeskorte et. al. analyzed neural recordings made in an anterior patch of macaque IT on a small number of widely varying naturalistic images of every-day objects, and additionally obtained fMRI recordings from the analogous region of human visual cortex [19]. These objects included human and animal faces and body parts, as well as a variety of natural and man-made inanimate objects. Three striking findings of this work were that (i) the population code (as measured by RDMs) of the macaque neurons strongly mirrors the structure present in the human fMRI data, (ii) this structure appears to be dominated by the separation of animate vs inanimate object classes (fig. 4B, lower right) and (iii) that none of a variety of computational models produced RDMs with this structure.

Individual unit neural response data from these experiments is not publicly available. However, we were able to obtain a set of approximately 1000 additional training images with roughly similar categorical distinctions to that of the original target images, including distributions of human and animal faces and body parts, and a variety of other non-animal objects [16]. We posited that the population code structure present in the anterior region of IT recorded in the original experiment is guided by functional goals similar to the task distinctions supported by this dataset. To test this, we computed linear classifiers from $N_{HMO}$ features for all the binary distinctions possible in the training set (e.g. "human/non-human", "animate/inanimate", "hand/non-hand", "bird/non-bird", &c). The linear weighting matrix $W$ derived from these linear classifiers was then used to produce an RDM matrix which could be compared to that measured originally. In fact, the HMO-based population RDM strongly qualitatively matches that of the monkey IT RDM and, to a significant but lesser extent, that of the human IT RDM (fig. 4B). This fit is significantly better than that of all models evaluated by Kriegeskorte, and approaches the human/monkey fit value itself (fig. 4A).

## 4    Discussion

High consistency with neural data at individual neuronal response and population code levels across several diverse datasets suggests that the HMO model is a good candidate model of the higher ventral stream processing. That fact that the model was optimized only for performance, and not directly for consistency with neural responses, highlights the power of functionally-driven computa-

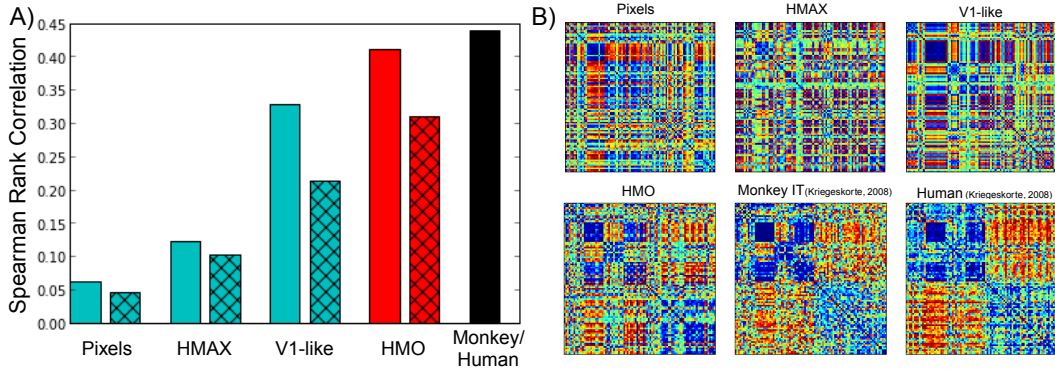

Figure 4: A) Comparison of model representations to Monkey IT (solid bars) and Human ventral stream (hatched bars). The HMO model followed by a simple task-blend based linear reweighting (red bars) quantitatively approximates the human/monkey fit value (black bar), and captures both monkey and human ventral stream structure more effectively than any of the large number of models shown in [18], or any of the additional comparison models we evaluated here (cyan bars). B) Representational Dissimilarity Matrices show a clear qualitative similarity between monkey IT and human IT on the one hand [19] and between these and the HMO model representation.

tional approaches in understanding cortical processing. These results further develop a long-standing functional hypothesis about the ventral visual stream, and show that more rigorous versions of its architecture and functional constraints can be leveraged using modern computational tools to expose the transformation of visual information in the ventral stream.

The picture that emerges is of a general-purpose object recognition architecture – approximated by the $N_{HMO}$ network – situated just posterior to a set of several downstream regions that can be thought of as specialized linear projections – the matrices $W$ – from the more general upstream region. These linear projections can, at least in some cases, be characterized effectively as the signature of interpretable functional tasks in which the system is thought to have gained expertise. This two-step arrangement makes sense if there is a core set of object recognition primitives that are comparatively difficult to discover, but which, once found, underlie many recognition tasks. The particular recognition tasks that the system learns to solve can all draw from this upstream "non-linear reservoir", creating downstream specialists that trade off generality for the ability to make more efficient judgements on new visual data relative to the particular problems on which they specialize. This hypothesis makes testable predictions about how monkey and human visual systems should both respond to certain real-time training interventions (e.g. the effects of "nurture"), while being circumscribed within a range of possible behaviors allowed by the (presumably) harder-to-change upstream network (e.g. the constraints of "nature"). It also suggests that it will be important to explore recent high-performing computer vision systems, e.g. [20], to determine whether these algorithms provide further insight into ventral stream mechanisms. Our results show that behaviorally-driven computational approaches have an important role in understanding the details of cortical processing[24]. This is a fruitful direction of future investigation for such models to engage with additional neural and behavior experiments.

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
