[Reviews · NeurIPS 2013]

Submitted by Assigned_Reviewer_1

Paper 1417 "Hierarchical Modular Optimization of Convolutional Networks Achieves Representations Similar to Macaque IT and Human Ventral Stream"
This paper describes a novel procedure for finding a set of heterogeneous neural response filters in a hierarchical model of the visual ventral stream. The main motivation for this work is that previous models fail to achieve the specific categorical structures present in neural responses in the top end of the stream, IT.

Quality. The results are strong and convincing, and the model demonstrates both a high similarity to neural IT responses, and demonstrates outstanding performance in 8-way classification tasks.

Clarity. The paper is reasonably well written, though highly condensed. Some elaboration on the HMO procedure would be welcomed, as it required multiple readings to understand (and there is space left in the paper).

Significance. The paper represents a substantial and convincing step in computational models of deeper brain structures.

It would have been interesting to see how the results scale with features derived from big data, like deep learning approaches.

Summary: The paper describes of novel and successful method for deriving the heterogeneity of feature detectors needed for building a deep model of the ventral stream, in particular for modeling object recognition area IT.

Submitted by Assigned_Reviewer_4

Summary:

This paper presents an algorithm for category-level object recognition that searches a large space of heterogeneous hierarchical neural network architectures using a boosting mechanism. The resulting network found via this algorithm is used to predict multiunit neural activity in monkey IT, as well as the similarity structure in monkey neural IT and human fMRI IT representations. This is the first reasonably successful prediction of neural activity in IT in response to complex naturalistic images of objects.

Pros:

Predicts neural activity in IT in response to complex naturalistic images of objects for the first time and with reasonable success. This is a significant achievement, and will be exciting to many at NIPS.

Introduces a new algorithm for category-level object recognition, which optimizes a very complex, heterogenous neural network architecture using a boosting methodology. This will be of interest also to non-neuroscientists interested in deep learning and object classification.

The comparisons to neural data are done to a high standard, with results replicated on three independently collected datasets (two monkey multiunit, one human fMRI).

The paper is clearly written in general.

Cons:

The Krizhevsky et al. object recognition network should be included in the evaluation, since this model was shown to be promising in closely related prior work (e.g., [2]). This is a very natural baseline model to compare to. The Krizhevsky network was shown to achieve classification performance superior to that of Monkey IT, on the exact Neural Representation Benchmark dataset used in this paper.

The paper describes running the proposed HMO algorithm to obtain a network N_{HMO} used in the comparisons to experimental data. It would be very interesting to learn more about the properties of this optimal network. What is its topology? Is the permitted heterogeneity in the space of networks used, or does this procedure end up using networks of some standard depth? Since intuition into what makes a good architecture is generally hard to come by, it would be useful to see this example. It would also be useful to describe other properties of the specific HMO implementation used in this paper, namely, how many networks were evaluated in the optimization of single stack networks, etc.

As it stands it is not possible to determine whether the proposed algorithm is necessary to model IT well, or whether other state-of-the-art object recognition methods would perform comparably. This has bearing on the view advanced at the end of the discussion that the paper's results argue for one relatively fixed, unmodifiable "core recognition" network which feeds into more plastic "downstream specialists" responsible for learning different tasks. The Krizhevsky network, by contrast, would be plastic throughout all levels, with no obvious "two-step" arrangement. If it performs comparably in predicting IT responses, it would weaken the paper's argument in support of their dual system view. On the other hand, if the Krizhevsky network underperformed the proposed method, this would be very informative. As it stands, arguing in support of the two-step arrangement seems premature given the other tenable but unexamined models in the literature.

Minor:

The description of the HMO algorithm is hard to follow. The score equation in line 143, Sum N(F(s)), makes sense to me as E(N(s)). Also it appears that F is never defined. The reweighted score function of line 156 again makes use of the undefined F, F_1, etc, which presumably again is N.

229: "Matricx" -> "Matrix"

Fig. 3A: include Y axis label
Summary: Predicts neural activity in IT in response to complex naturalistic images of objects for the first time and with reasonable success. Does not include an important baseline model.

Submitted by Assigned_Reviewer_5

This is an interesting paper proposing a hierarchical modular network of the ventral visual pathway which is learnt by boosting. This paper is written very clearly and comparison between model behavior and experimental data is also well done. Although I am not a specialist in the field, I feel the contribution of this paper important.
Summary: I would recommend the acceptance of the paper.

Submitted by Assigned_Reviewer_6

This paper presents a new computational model visual processing and shows that the the representation learned by this model more closely matches the response characteristics of IT cortex in the brain. The quality of the work is high, and I have no doubts about the validity of the results or the underlying methodology of experimentation.

The paper is original but not completely unprecedented. The HMO model itself is an interesting idea for constructing an ensemble model out many heterogeneous variants of a previously published visual model. The comparison of the hidden representation constructed by the model to real brain responses follows on some recent ideas in the literature, but the new model the authors suggest does seem to be an improvement compared to other published models evaluated on this metric.

The clarity of the paper is somewhat mixed. Some sections were exceptionally well written (e.g. the Introduction), while others were somewhat harder to follow than they might have been (e.g. High-Throughput Screening via Hierarchical Modular Optimization).

The work is significant principally because it is part of what seems like an important trend in literature: to develop quantitative models for visual computation and make meaningful comparisons between these models and the brain. In essence to treat computational models as well formed hypotheses for visual brain function (at least at the a course scale) and measure how well they account for real data.

Minor comments:
The section title "Intro" should be expanded to read "Introduction".
The word "nurture" is misspelled in the Discussion section.
Summary: This paper presents a new computational model visual processing and shows that the the representation learned by this model more closely matches the response characteristics of IT cortex in the brain.
Author Feedback

Author rebuttal: Thanks for the positive comments, those are always nice to get. We also basically agree with the substantive concerns raised. We don't view this as a rebuttal so much as a response lay out how we plan to address these concerns.

Here are our responses, to what we perceive as the main issues brought up in the reviews:

1) More sophisticated filterbank learning mechanisms: So far, we've chosen to use (uniform) random filters, and to control only the variance and means of these filters. From some parameter analysis that we've done, it seems that those gross filterbank statistics actually matter a lot, especially having multiple such values heterogenously composed in the modular components of the model. This suggests that the reviewer's suggestion could be quite important. Perhaps we could materially improve performance (and also neural fitting) by using more sophisticated mechanisms for choosing filterbank values, in additional to the architectural parameter optimization we have already worked on -- for example, the contrast filters described in the Ghebreab etal NIPS paper, or other approaches, like back-propagation and deep learning. We're addressing this now, for a future work, but probably won't have anything material to add in a revision. However, it is a good suggestion.

2) Comparison to some recent deep learning feature approaches, especially e.g. Krizhevsky et. al. This is a really important issue. One of the reviewers references our reference [2], a recent errfor lead by a collaborator in our group. There were some technical reasons we didn't include all the comparison feature sets compared to in reference [2]. First, of all we didn't actually have access to feature vectors from the algorithms evaluated in [2], for a significant subset of the images. That is, the "neural representation benchmark" (NRB) set that we used in this current work contains a significant number of images that were held out of the test in [2], including additional object categories. More important -- we didn't have features extracted for the algorithms in reference [2] for the Man Vs Monkey datasets (this especially goes to review #2's comments about the "two-step" model). As a result, it would have been hard to fit those algorithms in for a direct feature comparison, at least at the time of writing the paper. What we will try to do in preparing a final version (if accepted) -- is have those groups that participated in [2] extract features on the remaining images from the NRB dataset, as well as the Man Vs. Monkey datasets. We'll also want to do this with features from the algorithm described in recent work by Zeiler & Fergus.

What we have done already: we have the features from the algorithms compared in [2] for a subset of the NRB images. We have looked at the comparisons on that subset. The answer, that we can tell so far, is that the Krizhevsky features do pretty well at fitting the neural data -- somewhat less well overall than the HMO features we present here, but still, significantly better than the other control models. So we were happy about this, because it suggests further evidence for our main point, which is that by optimizing for performance on an object recognition task, one produces models that get better at predicting neural data. We suspect that once we get all the images [especially on the Man Vs. Monkey dataset] -- either as a revision for this paper, or perhaps (more likely) as a longer more detailed journal submission, we'll have a really strongly case for this, integrating these other new feature sets as data points.

Something interesting happens even with the data we do have now: in the course of the neurophysiology experiment, we measured data from neurons in two animals. The HMO features seem to do better at predicting one monkey, and the Krizhevsky features better at the other monkey. [It wasn't clear how to include this in in the existing paper, but maybe we should try to think of a way to do so in a revised version for final submission?] This made us wonder how well the neural features from one monkey predict the neurons from the other monkey -- and the answer is, basically no better than either the HMO or the Krizhevsky features. This has further begun to make us wonder, to what extent is the view of "IT" as a unified area in the brain really right? During the experiment, we placed arrays in slightly different parts of the two monkey's brains, both within anatomical IT, but still not exactly in the "same place" according to one or another coordinate systems (the utility of which are also suspect). This result lead us to wonder, is there substructure in IT that we can study and compare to models like HMO and SuperVision? This kind of result would be of real interest to neurophysiologists because it would suggest that we can make more detailed predictions about visual cortex structure by studying differences between models. In any case, this is a key next step that we are working on actively.

3) Reviewer # 2 is right to suggest that we did not record neurophysiology data for the purpose of this paper. We should make this more clear, and will do so. However, we want to be clear also that the neurophysiology data reported here is also NOT "taken" from our reference [2], as suggested. This neurophys data has actually not yet been reported anywhere in a journal paper! (Our lab submitted a paper last week with the main report on that data.) So both this current paper and [2] are derivative modeling efforts from core neurophysiology data developed and recorded by Ha Hong (one of the two first-authors of this current paper). We need to make that clear.

4) Typos in the description of the model will be fixed. Moreoever, and we'll try to make the text clearer; we've been working on text that is cleaner and simpler and we think will make the procedure more straightforward to understand.